# A Cost-Effective Lightning Current Measuring Instrument with Wide Current Range Detection Using Dual Signal Conditioning Circuits

**DOI:** 10.3390/s23063349

**Published:** 2023-03-22

**Authors:** Youngjun Lee, Young Sam Lee

**Affiliations:** 1Sungjin Techwin Co., Ltd., 62, Yuseong-daero 877 beon-gil, Yuseong-gu, Daejeon 34127, Republic of Korea; yjl0717@empal.com; 2Department of Electrical Engineering, Inha University, 100 Inha-ro, Michuhol-gu, Incheon 22212, Republic of Korea

**Keywords:** lightning current measuring instrument, lightning protection system, lightning monitoring system, lightning current signal processing

## Abstract

Lightning strikes can cause significant damage to critical infrastructure and pose a serious threat to public safety. To ensure the safety of facilities and investigate the causes of lightning accidents, we propose a cost-effective design method for a lightning current measuring instrument that uses a Rogowski coil and dual signal conditioning circuits to detect a wide range of lightning currents, ranging from hundreds of A to hundreds of kA. To implement the proposed lightning current measuring instrument, we design signal conditioning circuits and software capable of detecting and analyzing lightning currents from ±500 A to ±100 kA. By employing dual signal conditioning circuits, it offers the advantage of detecting a wide range of lightning currents compared to existing lightning current measuring instruments. The proposed instrument has the following features: First, the peak current, polarity, $T_{1}$ (front time), $T_{2}$ (time to half value), and Q (amount of energy of the lightning current) can be analyzed and measured with a fast sampling time of 380 ns. Second, it can distinguish whether a lightning current is induced or direct. Third, a built-in SD card is provided to save the detected lightning data. Finally, it provides Ethernet communication capability for remote monitoring. The performance of the proposed instrument is evaluated and validated by applying induced and direct lightning using a lightning current generator.

## 1. Introduction

Recently, owing to climate change and global warming, lightning has occurred more frequently than in previous years. The National Aeronautics and Space Administration reported that the incidence of lightning increases by approximately 5–6% when the global surface temperature increases by 1 °C [1,2]. Figure 1a shows the number of cloud-to-ground lightning strikes that occurred in Korea from 2012 to 2021, with an average of 115,609 occurrences per year. Figure 1b shows the number of cloud-to-ground lightning strikes per month in Korea in 2021. This indicates that lightning strikes frequently occur in summer when the weather is hot [3]. Lightning is a common natural phenomenon that primarily occurs under cloud-to-ground and cloud-to-cloud conditions. However, they cause damage to electrical installations, buildings, structures, airplanes, and humans because of their large currents and induced transient voltages [4,5,6,7]. To prevent such damage, the lightning protection system standards of the International Electrotechnical Committee (IEC) have been applied in most countries. The external lightning protection system consists of an air termination system such as a lightning rod, a down-conductor system, and an earth termination system. The internal lightning protection system consists of equipotential bonding and electrical separation from the external lightning protection equipment. In addition, surge protective devices are used to protect electrical equipment inside buildings [8,9,10,11,12,13]. The Seohae Bridge, a cable bridge in Korea, experienced a fire accident on 3 December 2015, resulting in social consequences and economic losses of approximately USD 30 million, including human accidents and traffic paralysis [14]. Figure 2 illustrates the lightning accident at the Seohae Bridge.

The accident investigation committee concluded that the accident was caused by a lightning strike. The Korean Ministry of Land, Infrastructure, and Transport decided to reinforce external lightning protection facilities and install a lightning current measuring instrument (LCMI) on all cable bridges to prevent lightning accidents, such as the fire on the Seohae Bridge. The primary functions of the LCMI required by the Korean Ministry of Land, Infrastructure, and Transport are as follows:Lightning current detection range is 500 A to 100 kA;The polarity of the lightning current should be distinguishable;The lightning current detection time and date must be known;Real-time remote monitoring should be provided through Ethernet communication;The product must have portable memory, and the lightning current detection data must be stored;International standard on waterproof and dustproof (IP code) is IP43 or higher.

Lightning accidents have occurred not only at the Seohae Bridge but also in other places [15,16,17,18].

However, only two LCMIs are available on the market. Table 1 compares the main features and prices of the LCMIs.

A CT or Rogowski coil is typically used as a sensor to measure lightning current, while recent technology has introduced the use of optical sensors or magnetostrictive composite sensors [7,19]. Although several papers on lightning current measuring instruments have been published, none have addressed a wide range of lightning current detection methods in measuring instruments. The product of P company could not be used because the current detection range did not meet these requirements. The product of I company is so large that installation space is limited. Moreover, its selling price is high because it utilizes an expensive current transformer (CT), leading to poor economic feasibility. Therefore, it is necessary to develop an LCMI with a reasonable price and easy-to-install size, and to study a wide range of lightning current detection methods. We developed an LCMI that can detect currents from ±500 A to ±100 kA and satisfied the above-mentioned requirements by designing a signal processing circuit and software running on a microcontroller. The proposed measuring instrument analyzes the lightning current signal with a fast sampling time of 380 ns of the microcontroller. Moreover, it extracts information about the peak current, amount of energy, $T_{1}$ (front time), $T_{2}$ (time to half value), polarity, and lightning class (whether lightning current is induced or direct). The performance of the proposed instrument is evaluated and validated by applying both induced and direct lightning using a lightning current generator.

## 2. Materials and Methods

### 2.1. Mechanical Design

The proposed LCMI is 140.0 × 117.0 × 85.5 mm, which is relatively small, and has an aluminum enclosure. External connectors for the AC/DC adapter, CT sensor, RS-232 communication, Ethernet, and SD card slot are placed on one side of the enclosure. The LCMI is designed to meet the IP64 for waterproof and dustproof ratings because it is installed in a coastal area. At the top of the LCMI are a liquid crystal display (LCD) and five touch keys. The product has four mounting holes on the left and right sides for fixation. Figure 3 shows the proposed LCMI.

### 2.2. Block Diagram

If a cable bridge is struck by lightning, the lightning current flows through the down conductor. A Rogowski coil with a passive RC integrator was installed to encircle the down conductor and sense the lightning current. The output of the Rogowski coil-based current sensor is fed into the signal conditioning circuit inside the LCMI. When a lightning current is detected, the signal conditioning circuit generates an interrupt signal and transfers the lightning current signal to the microcontroller. The microcontroller interrupt service routine analyzes the lightning current signal upon receiving the interrupt signal. The microcontroller measures $T_{1}$ (front time), $T_{2}$ (time to half value), Q (energy), and lightning class (whether the lightning current is induced or direct) through lightning current waveform analysis [7]. The microcontroller receives lightning current information analyzed by the microcontroller through a universal asynchronous receiver/transmitter (UART). It saves the lightning current information obtained from the DS and the date and time information from the real-time clock (RTC) to the SD card. For the battery, Seiko Semiconductor’s MS920SE-FL27E was used to prevent the date and time information of the RTC from being deleted when the power was turned off. The LCMI has an LCD to display information. The data are transmitted via Ethernet and RS-232C. The power circuit receives 15 VDC from the AC/DC adapter. The output of the power circuit consists of ±12 V for the signal conditioning circuit, +5 V for the microcontroller, and +3.3 V and +1.8 V for the microcontroller. Level shifters were used between the microcontroller and the microcontroller and between the microcontroller, SD card, and Ethernet drivers to adjust the voltage levels. The LCD was be used to check the detection data. A touch key was used for the field operation. Figure 4 shows a block diagram of the proposed LCMI.

The inside of the proposed LCMI consists of a main printed circuit board (PCB) and a key PCB. The PCBs are shown in Figure 5.

### 2.3. Lightning Current

#### 2.3.1. Classification of Lightning Current

Figure 6 shows the international standard current waveforms from direct and induced lightning, where $O_{1}$ is the virtual origin, I is the peak current, $T_{1}$ is the front time, and $T_{2}$ is the time to half the value. Table 2 presents a comparison of the characteristics of the two lightning current waveforms [8,11].

#### 2.3.2. Lightning Current Analysis

When induced lightning is detected on a cable bridge, the facility manager must check whether the electrical equipment is operating normally. In the case of direct lightning strikes, a high current flows through the lightning rods or horizontal conductors, and the facility manager must check whether there is any damage to the external structure. Therefore, if it is possible to distinguish between direct and induced lightning through an analysis of the lightning current, facility managers can manage lightning accidents more systematically. The IEC 62305-1 international standard specifies the lightning current parameters listed in Table 3 [7]. Direct lightning of negative polarity has a 95% probability that the Q value will be greater than 1.1 coulomb, where Q is the electric charge, which is the integrated value of the lightning current. It is expressed as
(1)Q=∫idt

The characteristics of the first positive short of a direct lightning strike have a 95% probability that the strike duration is over 25 μs. This value is distinguishable because it is longer than the time to half value (20 µs) of the induced lightning current. As a result, when the energy is over 1.1 coulomb and $T_{2}$ is over 25 µs, it can be roughly distinguished as a direct lightning strike.

### 2.4. Rogowski Coil Current Sensor and Signal Conditioning Circuit Design

#### 2.4.1. Rogowski Coil Current Sensor

In general, lightning current is detected using a current transformer. To measure a high surge current, a Rogowski coil that does not saturate even at a high current or an expensive and large current transformer with good frequency characteristics is used [20,21]. The proposed LCMI uses an air core Rogowski coil at a price of several hundred dollars, which is relatively economical. The voltage induced in the coil is proportional to the rate of change (differentiation) of the current flowing in the down conductor, and the lightning current signal is the output of the built-in passive RC integrator. The demountable Rogowski coil current sensor used in this study is shown in Figure 7, and its specifications are listed in Table 4 [22].

#### 2.4.2. A Signal Conditioning Circuit Capable of Detecting Lightning Current from 500 A to 100 kA

The signal conditioning circuit in Figure 8 consists of a voltage follower to remove the loading effect, an inverting amplifier composed of $U_{1A}$ that can process a lightning current signal of 500 A to 10 kA, and an inverting amplifier composed of $U_{1B}$ that can process a lightning current signal of 10 kA to 100 kA. In addition, the non-inverting amplifier configured in $U_{2B}$ can detect positive-polarity lightning currents, whereas the inverting amplifier configured in $U_{2A}$ can detect negative-polarity lightning currents. $D_{1}$, $D_{2}$, $D_{3}$, $D_{4}$, $ZD_{1}$, $ZD_{2}$, $ZD_{3}$, and $ZD_{4}$ are used as protection circuits to prevent the microcontroller overvoltage.

The output of the circular Rogowski coil follows Ampere’s circuit law, and the voltage ($e_{out}$) induced in the Rogowski coil can be expressed as [20]
(2)$$e_{out}=-\frac{d\Phi }{dt}=-M\frac{di}{dt}=\mu_{0}NA\frac{di}{dt},$$
where Φ is the magnetic flux, *M* is the mutual inductance, $\mu_{0}\;$ is the permeability of the vacuum, 4π × $10^{-7}$ m/H, *N* is the number of turns, and *A* is the cross-sectional area of the coil. The voltage induced in the Rogowski coil appears in the relationship between the differential value of the lightning current and the mutual inductance in the down conductor. Therefore, it should be compensated using an integrator [23,24]. The proposed LCMI lightning current sensor comprises RC passive integrators ($R_{2}$ and $C_{1}$) that do not require an external power source. A commercially available Rogowski coil is used, and the output of the sensor is approximately 50 mV at a lightning current of 500 A. Considering the frequency of the lightning current, THS4032 from Texas Instruments, an OP amp, is chosen for the signal conditioning circuit. Features of the THS4032 are:ultra-low 1.6 nV/√Hz voltage noise;high speed (100 V/μs slew rate);low 0.5 mV (typical) input offset voltage;100 MHz band width.

The lightning current signal is input into the analog-to-digital converter (ADC) of the microcontroller. A total of +3.3 V is applied to the ADC power of the microcontroller, and the input range is 0 V to 3.0 V. The reason why the bias voltage of $U_{1A}$ and $U_{1B}$ is designed as +1.5 V is to set it to about 50% of the allowable input voltage range of the microcontroller’s ADC so that the negative and positive lightning currents can be distinguished and analyzed. The inverting amplifier using $U_{1A}$ handles lightning current signals from 500 A to approximately 10 kA. The inverting amplifier using $U_{1B}$ handles lightning current signals from 10 kA to 100 kA. The output voltages of $U_{1A}$ and $U_{1B}$ can be calculated as follows:(3)$$V_{3}=-\frac{R_{6}}{R_{5}}V_{2},$$
(4)$$V_{4}=-\frac{R_{11}}{R_{10}}V_{2}.$$

The gain of the $U_{1A}$-based amplifier is −1, and the gain of the $U_{1B}$-based amplifier is −0.1. Table 5 lists the calculated voltages of $V_{2}$, $V_{3}$, and $V_{4}$ depending on the magnitude of the lightning current.

From the above table, it is seen that the signal becomes saturated at lightning current ±15 kA and ±100 kA. Therefore, considering the saturation margin, ADC0 is designed to detect 500 A to 10 kA and ADC1 is designed to detect 10 kA to 100 kA. The signal processing of the ADC, which depends on the magnitude of the lightning current, is shown in Figure 9.

The signal conditioning circuit proposed in this study was tested by applying an induced lightning current generated by a lightning current generator. The experimental setup is shown in Figure 10, the measurement equipment is listed in Table 6, and an example of the experimental results is shown in Figure 11.

In Figure 10, when an induced lightning current of 5 kA is applied from the lightning current generator, the output of the Rogowski coil appears as a differential waveform, the output $V_{2}$ of the voltage follower that has passed through the integrator appears similar to the applied current waveform, and the output voltage is measured as 550 mV. The waveform of $V_{3}$, which is an input to ADC0, appears as an inverted waveform. In addition, the waveform of $V_{4}$, input to ADC1, appears at 1/10th the amplitude of that of ADC0.

### 2.5. A Signal Processing Software Algorithm of the Microcontroller for Lightning Current

A TMS320F28335 from Texas Instruments was selected as the microcontroller for the proposed LCMI. The lightning current exhibits fast transient characteristics. Therefore, the ADC sampling time and resolution of the microcontroller are critical for accurate lightning current signal analysis. The ADC of the proposed LCMI has a sampling time of 380 ns and 12-bit resolution. The ADC sampling time performances of microcontrollers that are widely used in embedded systems are compared in Table 7 [25,26,27,28,29]. It is seen that the performance of TMS320F28335 is excellent.

When power is applied to the LCMI, the registers of the microcontroller are initialized. When a lightning current is detected, the converted data from ADC0 and ADC1 are stored in the memory through direct memory access (DMA). When the lightning current signal is terminated, the stored data of ADC0 and ADC1 are read back through the DMA. The reason for this process is to make the ADC sampling time as fast as possible. To remove the noise from the lighting current signal, signal processing is performed using a low pass filter (LPF). The microcontroller then calculates six parameters of the lightning current signal: $T_{1}$, $T_{2}$, Q (coulomb), peak current, polarity, and lightning class. If the peak current is less than 0.4 kA or the duration time of the lightning current is less than 5 μs, it is classified as noise. If the peak current of ADC1 is greater than 10 kA, 6 lightning parameters extracted from ADC1 are transmitted to the microcontroller through the UART. If the peak current of ADC1 is less than 10 kA, the 6 lightning parameters extracted from ADC0 are transmitted to the microcontroller through the UART. The signal processing software algorithm for the lightning current is shown in Figure 12.

### 2.6. A Software Algorithm of the Microcontroller (ATMEGA1280)

Figure 13 shows the software flowchart of the Microchip company’s ATMEGA1280 microcontroller, which is manufactured in Chandler, USA.

When power is applied to the microcontroller ATMEGA1280, it initializes the peripheral devices. The microcontroller reads the lightning current detection data from the SD card and the date and time information from the RTC. The lightning current detection and time information are displayed on the LCD. When the touch key is pressed, the LCD screen is updated accordingly. When an LCMI receives the lightning current detection information from the microcontroller (TMS320F28335), it displays the information on the LCD, stores the information in the SD card memory, and transmits the data through RS-422 and Ethernet communications for remote monitoring. The LCD information display consists of four screens. The first screen shows the number of direct and induced lightning current detections, and the second screen shows the detected date and time, lightning class, polarity, Q (coulomb), $T_{1}$, $T_{2}$, and peak current. The third screen sets the Ethernet IP address for remote monitoring, and the fourth screen sets the date and time and shows the RESET function to initialize the internally detected stored data and the software version. An example of an LCD screen is shown in Figure 14.

### 2.7. A Software of the Remote Monitoring

The LMCI data are sent to a monitoring center for remote monitoring, which enables workers in the monitoring center to systematically monitor the data of LCMIs located around the country. A screen capture of the monitoring program is shown in Figure 15.

## 3. Results

### 3.1. Test Results of Induced Lightning Current

The proposed LCMI was tested by a national accredited certification body. The peak current test results showed that it was not detected at ±250 A less than ±400 A and had a tolerance of about 10% in the range of 500 A to 80 kA. The test results of the lightning class were all correctly detected, and $T_{1}$, $T_{2}$ and Q were similar to those of the applied lightning current. The test results for the induced lightning currents are listed in Table 8.

### 3.2. Test Results of Direct Lightning Current

The proposed LCMI was tested by a national accredited certification body. The peak current test results showed that there is a maximum tolerance of about 10% in the test conditions of about ±50 kA and ±100 kA. The test results of the lightning class were all correctly detected, and $T_{1}$, $T_{2}$ and Q were similar to those of the applied lightning current. The test results for the direct lightning current are presented in Table 9.

## 4. Discussion

Table 10 compares performance and cost of LCMIs from different companies. The proposed LCMI is a relatively small product compared with that of I Company. To sense the lightning current, we used a Rogowski coil, which is relatively difficult to process but economical, instead of using an expensive CT. The lightning current range was the same as that of I Company, and the detection tolerance was approximately within 10%. A complete comparison is not possible because information on the detection tolerance is not listed in the product specifications of other companies. The lightning current polarity and maximum current were detected for all LCMI products. The proposed LCMI can detect the energy of the detected lightning current and the lightning class information, but P and I company products cannot detect it. Lightning risk assessment is performed every year based on lightning current detection data, and it has the advantage of being able to calculate lightning density using the lightning class information of direct lightning currents. The proposed LCMI and P company’s product are similar in price and are very economical compared to the price of I company’s product.

South Korea has a total of 76 cable bridges. The Ministry of Land, Infrastructure, and Transport plans to apply LCMIs to 56 cable bridges between 2019 and 2030. The proposed LCMI was installed and operated on eight cable bridges, as shown in Table 11 and Figure 16.

## 5. Conclusions

In this paper, we propose a design methodology for detecting lightning currents using a Rogowski coil that can detect a wide range of current magnitudes from ±500 A to ±100 kA. The methodology utilizes dual signal conditioning circuits to distribute the magnitude of the input signal and enable detection of a wide range of lightning currents through microcontroller signal processing and analysis. When using only one signal conditioning circuit, the current can only be detected within the range of ±500 A to ±10 kA. By employing dual signal conditioning circuits, the methodology offers the advantage of detecting a wide range of lightning currents. The microcontroller can extract information on six major parameters of the lightning current, including $T_{1}$ (front time), $T_{2}$ (time to half value), Q (Coulombs), peak current, polarity, and lightning class. To evaluate the proposed methodology, we tested and evaluated both induced and direct lightning currents using a lightning current generator, and found that the tolerance of the test results was within about 10%. The LCMI can be used to ensure the safety of important facilities and to determine the cause of lightning accidents, as well as to calculate the insulation distance using lightning current detection data.

## Figures and Tables

**Figure 1 sensors-23-03349-f001:**
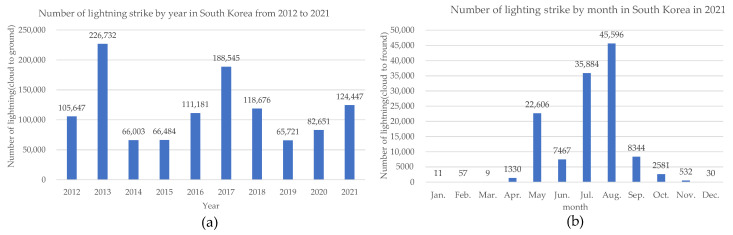
(**a**) Number of lightning strike by year in South Korea from 2012 to 2021. (**b**) Number of lightning strikes by month in South Korea in 2021.

**Figure 2 sensors-23-03349-f002:**
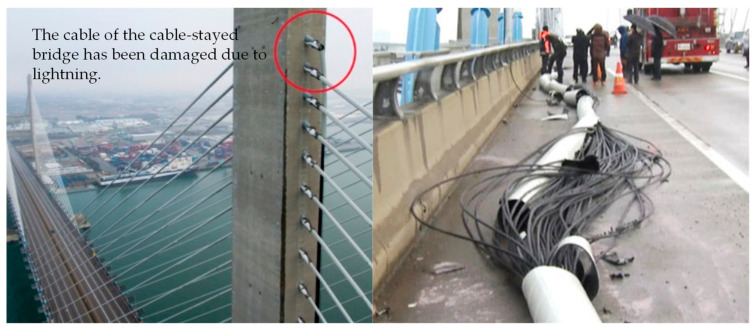
Lightning accident at the Seohae Bridge.

**Figure 3 sensors-23-03349-f003:**
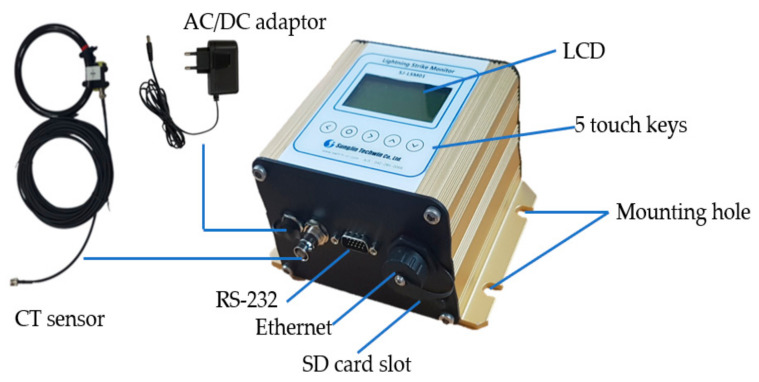
Shape of the proposed LCMI.

**Figure 4 sensors-23-03349-f004:**
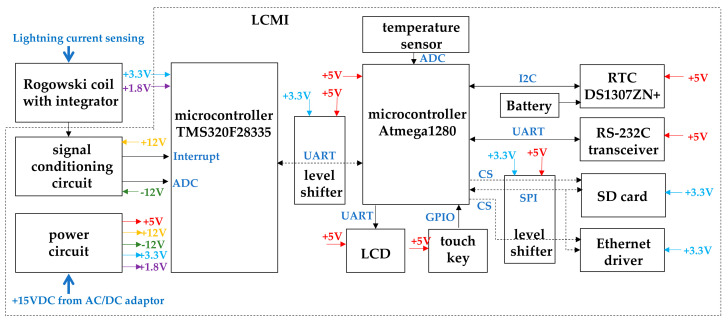
Block diagram of the proposed LCMI.

**Figure 5 sensors-23-03349-f005:**
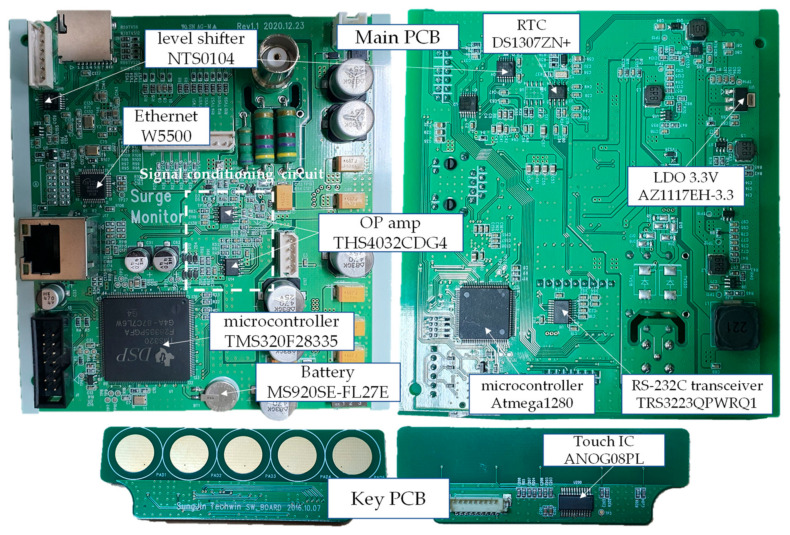
PCBs of the proposed LCMI.

**Figure 6 sensors-23-03349-f006:**
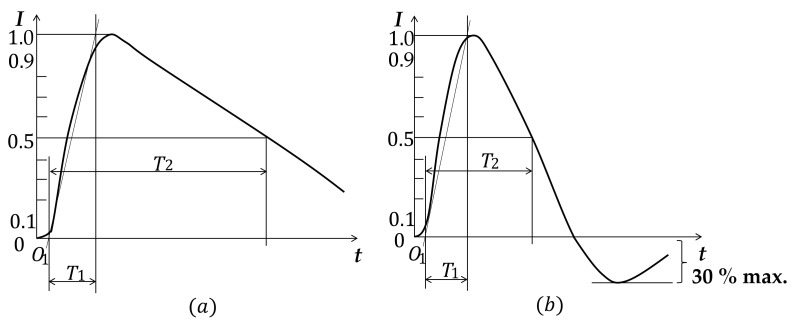
(**a**) Direct lightning current waveform; (**b**) induced lightning current waveform.

**Figure 7 sensors-23-03349-f007:**
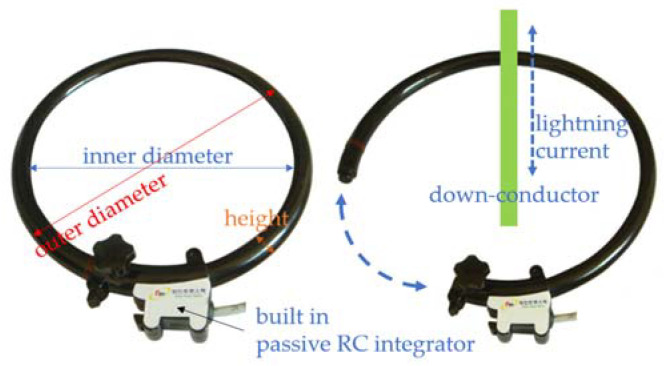
Shape of the Rogowski coil current sensor.

**Figure 8 sensors-23-03349-f008:**
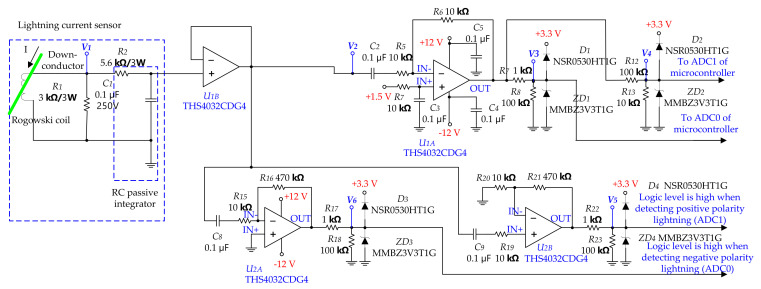
Dual signal conditioning circuits capable of processing of the lightning current from ±500 A to ±100 kA.

**Figure 9 sensors-23-03349-f009:**
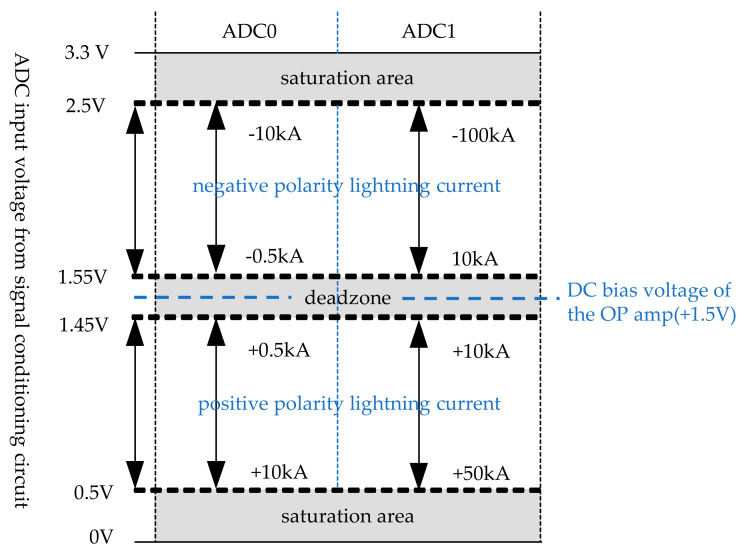
A signal processing diagram of the ADC depending on the magnitude of the lightning current.

**Figure 10 sensors-23-03349-f010:**
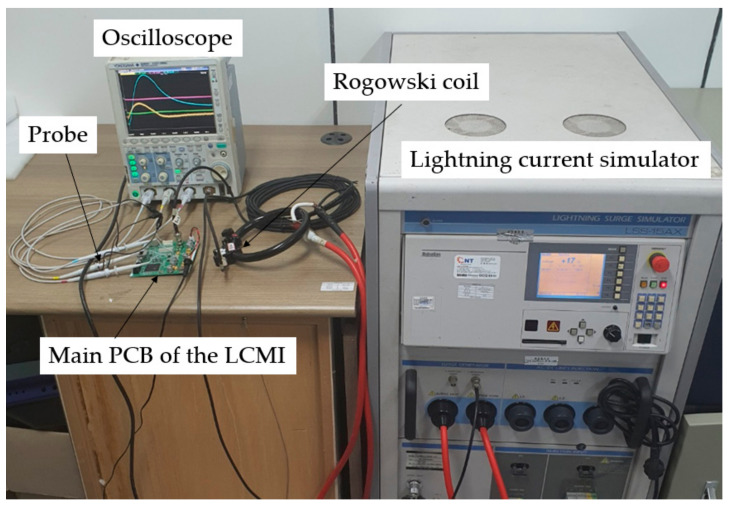
Experimental test setup for a signal conditioning circuit.

**Figure 11 sensors-23-03349-f011:**
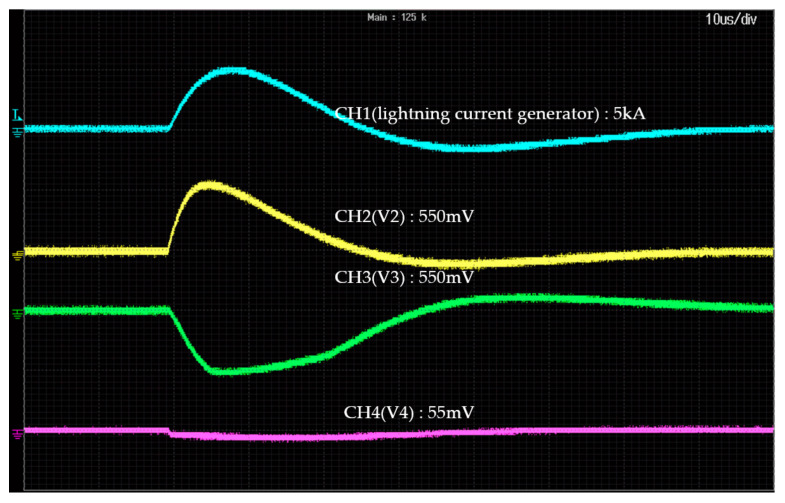
An example of the experimental results.

**Figure 12 sensors-23-03349-f012:**
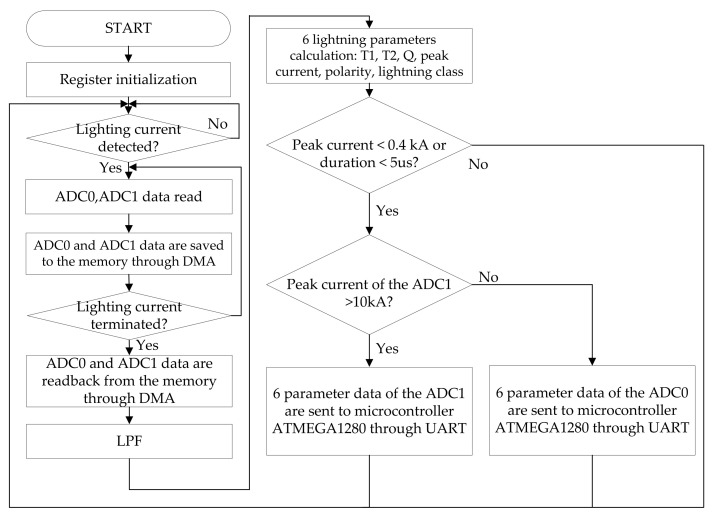
Signal processing software algorithm of the microcontroller for the lightning current.

**Figure 13 sensors-23-03349-f013:**
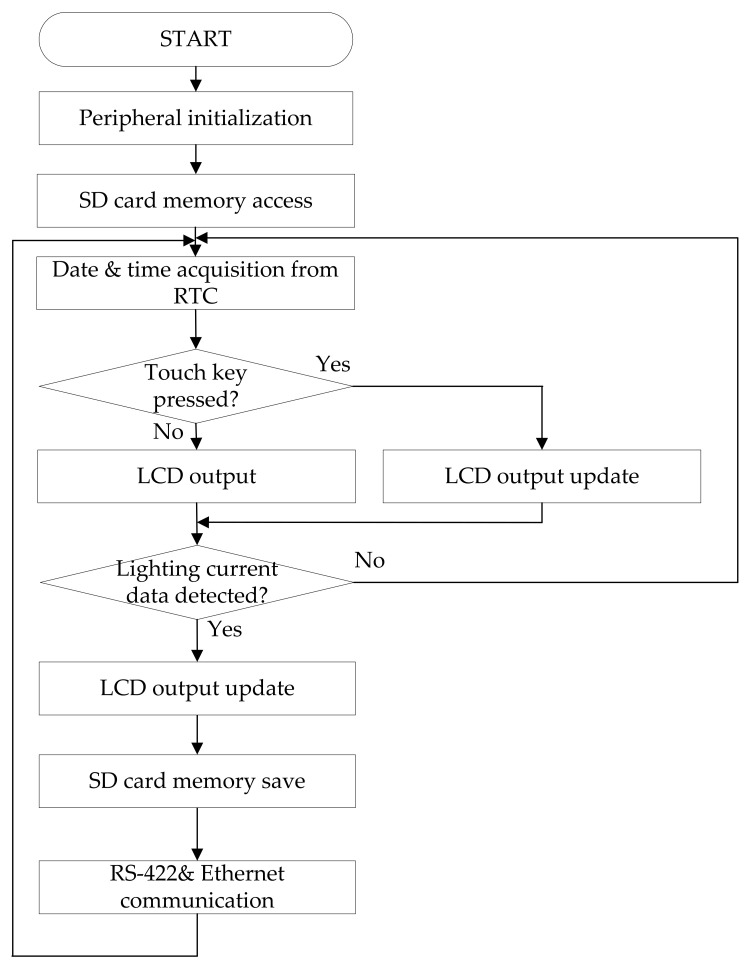
Software flowchart of the microcontroller ATMEGA1280.

**Figure 14 sensors-23-03349-f014:**
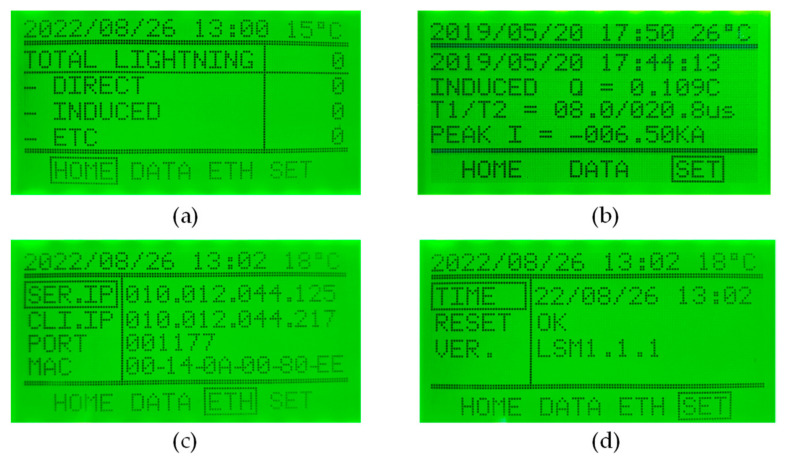
(**a**) First screen of the LCD. (**b**) Second screen of the LCD. (**c**) Third screen of the LCD. (**d**) Fourth screen of the LCD.

**Figure 15 sensors-23-03349-f015:**
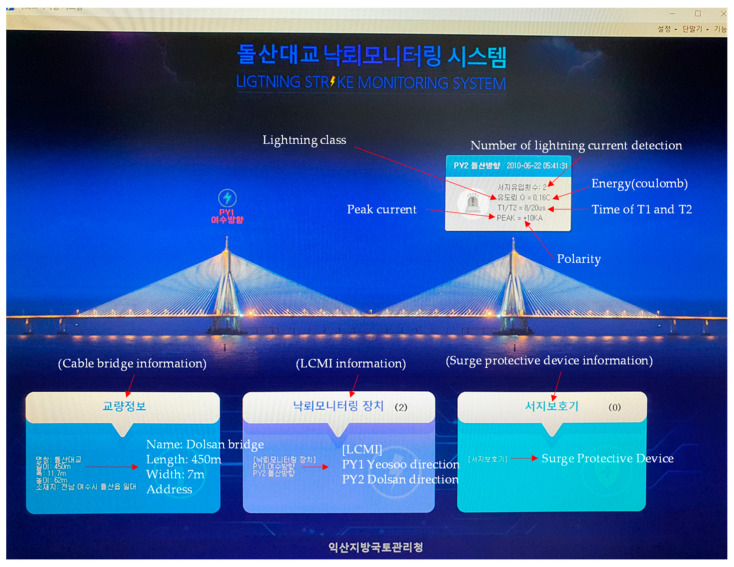
Screen capture of a remote monitoring program.

**Figure 16 sensors-23-03349-f016:**
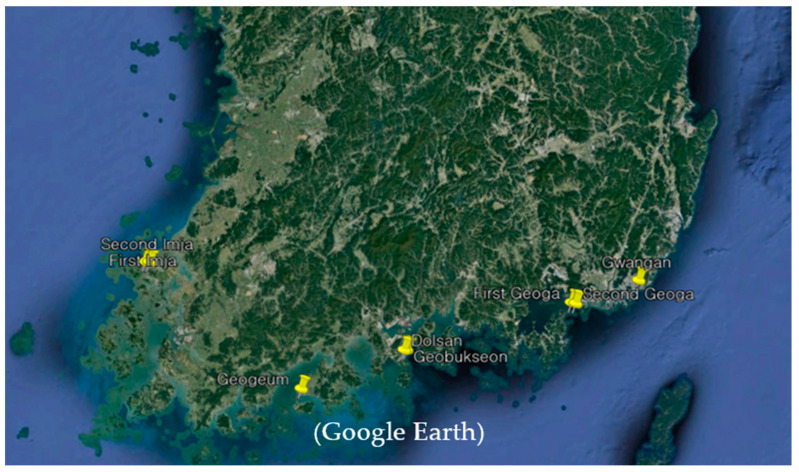
Place where the proposed LCMI is installed on the cable bridge.

**Table 1 sensors-23-03349-t001:** Comparison of the main features and prices of the LCMIs.

Features	P Company	I Company
Size (mm)	77.6 × 185 × 181.5	560 × 560 × 750
sensor	optical lightning sensor	CT (Current transformer)
current detection range (kA)	±5–400	±0.5–100
IP class	IP20	IP43
Internal memory	none	SD card
time and date information	yes	yes
price (USD)	few thousand	few tens of thousands

**Table 2 sensors-23-03349-t002:** Characteristics of lightning current waveforms.

Lightning Current	$T_{1}$ (µs)	$T_{2}$ (µs)
Direct	10	350
Induced	8	20

**Table 3 sensors-23-03349-t003:** Tabulated values of lightning current parameters taken from CIGRE.

Parameter	Values			Type of Strike
	95%	50%	5%	
Q (C)	1.1	4.5	20	Negative flash
	2	16	150	Positive flash
Strike duration (µs)	30	75	200	First negative short
	25	230	2000	First positive short (single)

**Table 4 sensors-23-03349-t004:** Specification of the Rogowski coil current sensor.

Part No.	Inner Diameter	Outer Diameter	Height	Current Detection Range	Frequency Range
FR-450	158 mm	126 mm	16 mm	1 A–100 kA	10 Hz–500 kHz

**Table 5 sensors-23-03349-t005:** Calculated voltages of $V_{2}$, $V_{3}$, and $V_{4}$ depending on the magnitude of the lightning current.

Lightning Current (kA)	$V_{2}$ $\;(V_{\mathrm{AC}})$	$V_{3}$$\;(V_{\mathrm{DC}\;\mathrm{bias}+\mathrm{AC}})$; ADC0	$V_{4}$$\;(V_{\mathrm{DC}\;\mathrm{bias}+\mathrm{AC}})$; ADC1
none	0	1.5	1.5
+0.5	−0.05	1.5 − 0.05	1.5 − 0.005
−0.5	−0.05	1.5 − 0.05	1.5 − 0.005
+10	+1.0	1.5 + 1.0	1.5 + 0.1
−10	−1.0	1.5 − 1.0	1.5 − 0.1
+15	+1.5	1.5 + 1.5 (signal saturation)	1.5 + 0.15
−15	−1.5	1.5 − 1.5 (signal saturation)	1.5 − 0.15
+100	+10.0	1.5 + 10.0 (signal saturation)	1.5 + 1.0
−100	−10.0	1.5 − 10.0 (signal saturation)	1.5 − 1.0

**Table 6 sensors-23-03349-t006:** Test equipment to measure the signal conditioning circuit of the LCMI.

Test Equipment	Maker	Model	Specification
Lightning current generator	Noiseken (Kanagawa, Japan)	LSS-15AX	*Voc* (1.2/50 µs): 15 kV *Isc* (8/20 µs): 7.5 kA
Oscilloscope	Yokogawa (Tokyo, Japan)	DLM2054	2.5 Gs, 500 MHz
Scope probe	Yokogawa (Tokyo, Japan)	701939	600 V, 600 MHz

**Table 7 sensors-23-03349-t007:** Performance comparison of ADC sampling time of microcontroller.

Part Name of the Microcontroller	Maker	Maximum Resolution	ADC Sampling Time
TMS320F28335	Texas Instrument (Dollas, USA)	12-bits	380 ns
ATMEGA1280	Microchip (Chandler, USA)	10-bits	13–260 μs
SAM3X/SAM3A series	Microchip (Chandler, USA)	12-bits	1 μs
STM32MP157	STMicroelectronics (Geneva, Switzerland)	16-bits	222 ns @ 12-bit
dsPIC33EP	Microchip (Chandler, USA)	12-bits	910 ns

**Table 8 sensors-23-03349-t008:** Test results of induced lightning current.

Applied Lightning Current		Test Results				
(8/20 µs)	Q (coulomb)	Peak Current	Tolerance (%)	Lightning Class	$T_{1}$$/T_{2}$ (µs)	Q (coulomb)
+0.25 kA	0.0047	Not detected	-	-	-	-
−0.25 kA	−0.0047	Not detected	-	-	-	-
+0.5 kA	0.0095	+0.53	6	Induced	7.5/21.1	0.011
−0.5 kA	0.0095	−0.48	4	Induced	8.3/20.5	0.009
+10.2 kA	0.1932	+10.9	6.8	Induced	7.9/21.2	0.209
−9.9 kA	0.1876	−9.7	2.1	Induced	8.7/21.6	0.186
+40.5 kA	0.7674	+42.5	4.9	Induced	7.9/21.1	0.818
−39.7 kA	0.7523	−38.4	3.3	Induced	7.5/21.3	0.739
+81 kA	1.534	87.3	7.3	Induced	7.4/22.0	1.680
−79.2 kA	1.5	72.3	8.7	Induced	7.9/21.8	1.391

**Table 9 sensors-23-03349-t009:** Test results of direct lightning current.

Applied Lightning Current		Test Results				
(10/350 µs)	Q (coulomb)	Peak Current	Tolerance (%)	Lightning Class	$T_{1}$$/T_{2}$ (µs)	Q (coulomb)
+54.0 kA	27.1	+52.16 kA	3.4	Direct	11.1/342.4	26.2
−51.2 kA	25.7	−49.77 kA	2.8	Direct	8.5/333.7	23.4
+103.2 kA	51.8	+113.05 kA	9.5	Direct	10.5/320.2	49.3
−104.0 kA	52.1	−100.8 kA	3.1	Direct	9.7/310.7	48.3

**Table 10 sensors-23-03349-t010:** Performance and cost comparison of the LCMIs.

LCMI	Size (mm)	Detection Range (kA)	Sensor	Polarity	Energy	Peak Current	Lightning Class	Cost (USD)
Proposed LCMI	140 × 117 × 85.5	±0.5–100	Rogowski coil	O	O	O	O	Few thousand
P Company LCMI	77.6 × 185 × 181.5	±5–400	Fiber optic	O	X	O	X	Few thousand
I Company LCMI	560 × 560 × 750	±0.5–100	CT	O	X	O	X	Few tens of thousands

**Table 11 sensors-23-03349-t011:** Cable bridge installation status of the proposed LCMI.

Name of the Cable Bridges	Location (Latitude, Longitude)	Installation Amount	Year of Installation
Geobukseon	34.7344° N, 127.7489° E	2	2019
Geogeum	34.5001° N, 127.1282° E	2	2019
Gwangan	35.1477° N, 129.1300° E	2	2020
Dolsan	34.7307° N, 127.7345° E	2	2021
First Imja	35.0863° N, 126.1259° E	2	2021
Second Imja	35.0865° N, 126.1517° E	2	2021
First Geoga	35.0176° N, 128.7288° E	3	2022
Second Geoga	35.0151° N, 128.7545° E	2	2022

## Data Availability

The datasets generated during and/or analyzed during the current study are available from the corresponding author upon reasonable request.

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
