# Peer review of "A Cost-Effective Lightning Current Measuring Instrument with Wide Current Range Detection Using Dual Signal Conditioning Circuits"

_sensors, 2023, doi:10.3390/s23063349_

Round 1

Reviewer 1 Report

Thank you for a nice design. It would be nice to see the device tested in a high voltage lab with much higher peak currents.

1. What is the main question addressed by the research?   The paper does not really address a question, it is a design for measuring impulse currents.   2. Do you consider the topic original or relevant in the field, and if so, why?   It is not original. It is only relevant in that it provides details of an implementation, but there is no new knowledge here.   3. What does it add to the subject area compared with other published material?   A different approach to a design.   4. What specific improvements could the authors consider regarding the methodology?   The methodology is fine for a design paper.   5. Are the conclusions consistent with the evidence and arguments presented and do they address the main question posed?   Yes.   6. Are the references appropriate?   Yes.    7. Please include any additional comments on the tables and figures.   The tables and figures are clear and appropriate.

Author Response

  1. What is the main question addressed by the research? The paper does not really address a question, it is a design for measuring impulse currents. 
    => This paper presents a design methodology for measuring a wide range of impulse current signals. Specifically, it describes a signal processing circuit and software processing method that utilizes multiple ADC channels to automatically select the appropriate ADC according to the magnitude of lightning current and process the signal to measure a wide range of current signals
  2. Do you consider the topic original or relevant in the field, and if so, why?   It is not original. It is only relevant in that it provides details of an implementation, but there is no new knowledge here
    => We believe that the topic of this paper is both original and highly relevant to the field. While measuring lightning current using a Rogowski coil is a common method, it is difficult to measure lightning current over a wide range (such as ±500A to ±100kA) using a general design method with DSP. To address this challenge, we propose a novel signal conditioning circuit and software algorithm for processing lightning current signals. Specifically, our proposed method utilizes ADCs to receive and analyze the lightning current signal through DSP signal processing, enabling the detection of a wide range of lightning current signals
  3. What does it add to the subject area compared with other published material?   A different approach to a design
  4. What specific improvements could the authors consider regarding the methodology?   The methodology is fine for a design paper
    => While measuring lightning current using a Rogowski coil is a common method, it is not feasible to measure lightning current over a wide range (such as ±500A to ±100kA) using a general design method with DSP. In this paper, we propose a novel approach for addressing this limitation by introducing a signal conditioning circuit and software algorithm for processing lightning current signals. Specifically, our proposed method utilizes ADCs to receive and analyze the lightning current signal through DSP signal processing, allowing for the detection of a wider range of lightning current signals
  5. Are the conclusions consistent with the evidence and arguments presented and do they address the main question posed?   Yes
    => The evidence and arguments presented in the conclusion of the paper are coherent, and the conclusion sentence has been enhanced. 
  6. Are the references appropriate?   Yes
    => We added eleven reference in the revised paper.
  7. Please include any additional comments on the tables and figures.   The tables and figures are clear and appropriate
    => We replaced Figure 6 so that it has better clarity in letters and numbers 

Reviewer 2 Report

Review

In this paper is presented a lightning current measuring instrument using a digital signal processor, and software, which is capable of measurements from hundreds of A up to hundreds of kA. The instrument has Ethernet communication for remote monitoring, has a built-in SD card to save the measured data, uses a fast-sampling time of 380 ns and detects whether the current is induced, or direct. This instrument can be used in lightning facilities for safety purposes and to study the causes of lightning accidents.

Is this reviewer opinion that the paper is interesting and describe useful instrumentation for lightning detection and measurements. The manuscript is well structured, contains development, implementation and experimental results.

The State of the art and the related References could be enriched, to address both aspects: a) the problem of lightning accidents and b) the measuring instrumentation.

Figure 6 must improve clarity of letters and numbers.

Some of the components and microcontrollers used in the implementation of the experimental system (such as in Table 7) must be referenced in the manuscript and added in the list of References.

Author Response

The State of the art and the related References could be enriched, to address both aspects: a) the problem of lightning accidents and b) the measuring instrumentation.

=> We have added the latest technology and references regarding lightning current detection sensors 

Figure 6 must improve clarity of letters and numbers.

=> We replaced Figure 6 so that it has better clarity in letters and numbers 

Some of the components and microcontrollers used in the implementation of the experimental system (such as in Table 7) must be referenced in the manuscript and added in the list of References

=> Related references have been newly added in the revised paper. 

Reviewer 3 Report

Put more and interesting experimental results.

Improve the quality of the figures.

Improve the conclusions.

Author Response

Put more and interesting experimental results.

=> We believe that the experimental results presented in this paper represent the essential findings. Therefore, it may be challenging to include more intriguing experimental results in this paper. 

Improve the quality of the figures

=> We replaced Figure 6 so that it has better clarity in letters and numbers.

Improve the conclusions.

=> We have rewritten the Conclusions section to more effectively convey the main idea and results of the paper.